# Short-Term Efficacy and Safety of Cataract Surgery Combined with Iris-Fixated Phakic Intraocular Lens Explantation: A Multicentre Study

**DOI:** 10.3390/jcm10163672

**Published:** 2021-08-19

**Authors:** Miki Kamikawatoko Omoto, Hidemasa Torii, Sachiko Masui, Masahiko Ayaki, Ikuko Toda, Hiroyuki Arai, Tomoaki Nakamura, Kazuo Tsubota, Kazuno Negishi

**Affiliations:** 1Department of Ophthalmology, Keio University School of Medicine, Tokyo 160-8582, Japanhidemasatorii@yahoo.co.jp (H.T.); m.sac@a7.keio.jp (S.M.); mayaki@olive.ocn.ne.jp (M.A.); tsubota@z3.keio.jp (K.T.); 2Minamiaoyama Eye Clinic, Tokyo 107-0061, Japan; toda@minamiaoyama.or.jp; 3Minatomirai Eye Clinic, Kanagawa 220-6208, Japan; arai@minatomiraieye.jp; 4Queen’s Eye Clinic, Kanagawa 220-6204, Japan; 5Nagoya Eye Clinic, Aichi 456-0003, Japan; nic@bc5.so-net.ne.jp

**Keywords:** cataract, phakic intraocular lens, multicentre study

## Abstract

The purpose of this study was to evaluate the short-term efficacy and safety of cataract surgery for patients with iris-fixated phakic intraocular lenses (pIOLs). This study included 96 eyes of 91 patients. The changes in the logMAR uncorrected visual acuity (UCVA), best-corrected visual acuity (BCVA), subjective spherical equivalent (SE), astigmatism, and endothelial cell density (ECD) were collected retrospectively. The intraoperative and postoperative complications also were investigated to assess the surgical safety. The preoperative UCVA and BCVA improved significantly at month 1 postoperatively, respectively (*p* < 0.001 for both comparisons). The efficacy and safety index at month 1 postoperatively were 1.02 ± 0.56 and 1.31 ± 0.64, respectively. The SE at month 1 postoperatively was significantly (*p* < 0.001) higher compared to preoperatively, whereas the subjective astigmatism did not differ significantly (*p* = 0.078). The ECD significantly decreased at month 1 (*p* < 0.001). The most common postoperative complication was intraocular pressure elevation exceeding 25 mmHg in 10.4% of eyes, which was controlled with medications in all cases until month 1 postoperatively. No intraoperative complications developed. Cataract surgeries for patients with iris-fixated pIOLs were performed safely with good visual outcomes.

## 1. Introduction

Uncorrected refractive error is a major cause of visual impairment worldwide [1], and the prevalence of myopia is reported to be growing, especially in Asian countries [2,3,4]. Implantation of phakic intraocular lens (pIOL) is an option to correct myopia [5,6]. The reversibility when necessary should be an advantage of pIOL implantation compared to laser corneal refractive surgery, such as laser in situ keratomileusis (LASIK). Some studies have reported good long-term outcomes up to 10 years [7,8,9]. However, some cases need pIOL explantation due to cataract formation or decreased endothelial cell density (ECD) [10,11,12]. Some studies have reported the safety and efficacy of combined cataract surgery/pIOL explantation; however, small case series [13,14,15,16,17] or case reports of new surgical techniques [18,19], except for the study by Vargas et al., investigated 87 eyes of 55 patients [20]. Furthermore, including the study of Vargas et al., most of these studies focused on posterior-chamber pIOLs. Anterior-chamber pIOLs are associated with a lower rate of cataract formation and pigment dispersion compared to posterior-chamber pIOL [5,21]. However, few studies have investigated pIOL explantation and cataract surgeries for eyes with iris-fixated pIOL. We report the short-term efficacy and safety of cataract surgery with iris-fixated pIOL explantation.

## 2. Materials and Methods

### 2.1. Study Institutions and Institutional Review Board Approval

This was a multicentre (Keio University Hospital, Minamiaoyama Eye Clinic, Minatomirai Eye Clinic, Queen’s Eye Clinic, and Nagoya Eye Clinic), retrospective, observational study. The Research Ethics Committee of the Keio University School of Medicine (approval number: 20190278) approved the study, and the other eye clinics participating in the study were described as collaborators in the ethics committee document and were thus covered under the approval granted by the Keio University School of Medicine. This study was conducted according to the tenets of the Declaration of Helsinki. Patients or the public were not involved in the design, conduct, reporting, or dissemination plans of our research.

### 2.2. Participants

One hundred and fifty-nine eyes of 139 patients were enrolled in the study; all had undergone pIOL explantation followed by phacoemulsification and IOL implantation at one of the five hospitals between December 2010 and April 2020. The inclusion criteria were eyes with an iris-fixated pIOL. The exclusion criteria were eyes with a vision-threatening disease except cataract, i.e., keratoconus, retinal detachment, central serous chorioretinopathy, macular edema, glaucoma, and choroidal neovascularization; or eyes that had undergone a previous ophthalmic surgery except pIOL implantation, i.e., LASIK, vitrectomy, and glaucoma surgeries. Therefore, 96 eyes of 91 patients were included in the final analysis.

### 2.3. Surgical Technique

Five surgeons performed all of the surgeries. A pIOL was explanted through a temporal or superior sclerocorneal incision (range, 2.4–7.0 mm), the size of which was determined based on the material from which the implanted pIOL was made, i.e., polymethyl methacrylate (PMMA) (Artisan^®^ or Artisan Toric^®^, Ophtec BV, Groningen, The Netherlands) or silicone (Artiflex^®^, Ophtec BV). The nylon suture was set when the PMMA lens was explanted, which was left at the site until the end of the study period. Standard phacoemulsification and IOL implantation then were performed through a temporal or superior corneal incision (range, 2.3–2.4 mm). The surgeon chose the type of IOL based on the patient’s request. The implanted IOLs are summarized in Appendix A. The IOL power was calculated using Barrett Universal II Formula with the preoperative measurements of axial length, keratometry, and anterior chamber depth. The anterior chamber depth was manually measured and verified for accuracy because the participants had pIOLs. A topical antibiotic (moxifloxacin hydrochloride) and a corticosteroid (betamethasone sodium phosphate) were administered 3 times daily for one week and a non-steroidal anti-inflammatory agent (diclofenac sodium) for 3 months postoperatively. Drug doses were tapered over the postoperative course.

### 2.4. Ophthalmologic Examinations

The uncorrected visual acuity (UCVA) was measured preoperatively and on day 1, week 1, and month 1 postoperatively. The best-corrected VA (BCVA) was measured at the same time points; however, in about half of the cases, this examination was omitted on postoperative day 1. These VAs were calculated in logarithm of the minimum angle of resolution (logMAR) units. The subjective spherical equivalent (SE) and astigmatism were also collected at the same time points. The safety and efficacy index were calculated as the month 1 postoperative BCVA/preoperative BCVA and postoperative UCVA/preoperative BCVA. We calculated these indices because the current surgeries reported in this study were performed on patients without visual impairment in many cases. The decimal VA was used only for these calculations. The ECD was measured preoperatively and month 1 postoperatively using a specular microscope (EM-3000 (TOMEY, Tokyo, Japan) and CellChek SL, Noncon Robo II, or XII (Konan Medical, Hyogo, Japan). The axial length was measured using the IOLMaster 500 or IOLMaster 700 (Carl Zeiss Meditec AG, Jena, Germany).

### 2.5. Statistical Analysis

To reduce the possible bias of including both eyes of a patient, the values between the baseline and each time point were compared using a linear mixed model in which the random effect was the subjects. The linear mixed model adjusts for the hierarchical structure of the data, modeling in a way in which measurements are grouped within subjects [22,23]. This was followed by Dunnett’s test for multiple comparisons when comparing the values between the baseline and each time point [24]. Statistical significance was set at 0.05. All analyses were performed using R 4.0.4 (R Foundation for Statistical Computing, Vienna, Austria).

## 3. Results

The mean ± standard deviation age of the patients at the time of cataract surgery was 55.0 ± 7.5 years. The duration between the cataract surgery and pIOL implantation was 9.7 ± 3.6 years. Fifty-three eyes received a PMMA phakic IOL and 43 eyes a silicone IOL. The UCVA and BCVA before the cataract surgery were 0.29 ± 0.34 and −0.01 ± 0.17 logMAR, respectively. The ECD was 1,986 ± 732 cells/mm^2^. The detailed information is summarized in Table 1.

Figure 1A and Table 2 show the changes in the UCVA. The preoperative value significantly improved at day 1, week 1, and month 1 postoperatively (*p* < 0.001 for all comparisons by a linear mixed effect model followed by Dunnett’s test). Similarly, the postoperative BCVA improved significantly at week 1 and month 1 (*p* < 0.001 for both comparisons) but not on day 1 (*p* = 1.0, Figure 1B, Table 2). The efficacy and safety indices on postoperative month 1 were 1.02 ± 0.56 and 1.31 ± 0.64, respectively.

The subjective SE was significantly larger (closer to 0) at all time points (*p* < 0.001 for all comparisons) (Figure 1C, Table 2), whereas the subjective astigmatism was greater on day 1 (*p* = 0.0060) but did not differ significantly at week 1 and month (*p* = 1.0 and *p* = 0.078, respectively) (Figure 1D, Table 2). The preoperative subjective astigmatism was significantly different between the patients with PMMA pIOLs and those with silicone IOLs (*p* = 0.022, Appendix A). However, this difference was not found postoperatively. The ECD significantly decreased at month 1 (*p* < 0.001) (Figure 1E, Table 2).

The most common postoperative complication was an intraocular pressure (IOP) elevation exceeding 25 mmHg, which occurred in 10.4% of cases on day 1. With the exception of one case, no elevations were observed at postoperative week 1. Including this case, the IOP of all cases were controlled with medications. Corneal edema was observed in 8.3% of cases on day 1, which were not observed on day 7. No intraoperative complications developed. Other postoperative complications are summarized in Table 3.

## 4. Discussion

In the current study, the efficacy and safety of cataract surgery combined with pIOL explantation were investigated in 96 eyes of 91 patients with an iris-fixated pIOL. This study included the largest number of cataract surgeries with pIOL explantation and was the largest study investigating patients with a iris-fixated pIOL. In a previous study with fewer cases, de Vries et al. [14] reported that the BCVA improved from 0.21 ± 0.21 to 0.17 ± 0.18. In the current study, the BCVA improved significantly from −0.01 ± 0.17 to −0.09 ± 0.10. A simple comparison of the studies was not possible because the baseline values differed in the study of de Vries et al., which included eyes with vision-threatening diseases, such as retinal detachment or myopic degeneration of the posterior pole. We excluded vision-threatening diseases; however, the improvements in the UCVA and BCVA were significant, with favorable efficacy and safety indices (1.02 ± 0.56 and 1.31 ± 0.64, respectively) at postoperative month 1, in light of the refractive correction.

In the current study, 10.4% of cases had a postoperative IOP elevation, despite the exclusion of glaucomatous eyes. A recent study by Vargas et al. [20] that included 87 eyes did not report IOP elevations. Meire et al. reported that two of 38 cases had ocular hypertension [16], one of which with steroid-induced ocular hypertension resulted in the need for an additional trabeculectomy because the steroids could not be discontinued due to systemic oncologic treatment. This rate was relatively high compared to uncomplicated cataract surgeries [25,26,27]. The exact reason is unclear; however, more intense inflammation that resulted from iris manipulation to remove the pIOL may be a reason. In the current study, the IOPs of all the cases were controlled safely only with medications until postoperative month 1; however, surgeons must be alert to IOP elevation postoperatively.

The current study had some limitations, one of which was the absence of a control group. Considering the baseline ECD (1986 ± 732 cells/mm^2^) with an average patient age of 55.0 ± 7.5 years, the corneal endothelial damage was probably an important reason for the surgery. Therefore, other surgeries, such as standard cataract surgery or cataract surgeries for eyes with a posterior-chamber pIOL, were not considered as suitable controls because the indications differed. Despite this, we believe our data, comprised of the largest sample size of cataract surgery for eyes with pIOL, are valuable.

In the current study, the UCVA improved significantly from 0.29 ± 0.34 to 0.03 ± 0.19 at postoperative at month 1. The value at day 1 (0.11 ± 0.30) improved significantly from the preoperative level. However, the differences between the targeted and postoperative refractive errors were not assessed in this study. Although it was reported that the preoperative biometric measures were generally accurate [28], some miscalculations in the axial length were found along with the subsequent hyperopic change [29]. Furthermore, the types of inserted lens varied and included toric and multifocal IOLs because of the multicentre study. The targeted refractive error in most current cases was emmetropia or weak myopia (mean targeted refractive error, −0.17 ± 0.49) and the postoperative SE was −0.17 ± 0.84. Therefore, satisfactory outcomes were achieved in most cases; however, the specific analysis, such as the optimal IOL calculation formula to be used, will be addressed in our next study.

Our follow-up period was short. Although the recovery from the surgery was favorable despite this short follow-up period, the information about the clinical outcomes and safety with longer follow-up is essential for clinicians. In particular, the ECD significantly decreased at 1 month after surgery. The explantation of pIOL was carefully performed through sclerocorneal incision in order to not touch the endothelium. This procedure specific to the surgery might have had an effect. However, the ordinary cataract surgery with phacoemulsification and IOL implantation is well known to have an effect on ECD. The ECD change in our study was, on average, 4.5%. This was comparable to the past study of ordinary cataract surgery [30,31]. Thus, the ECD change was relatively small, but the early endothelial cell change cannot be fully evaluated by ECD [32]. Although the number of cases will be limited due to the retrospective design, careful and longer follow-up is needed. This will be discussed in the near future.

In conclusion, cataract surgeries for patients with iris-fixated pIOL were performed safely with good visual outcomes. We believe this option may be considered for patients with a pIOL who have visual impairment and endothelial cell loss.

## Figures and Tables

**Figure 1 jcm-10-03672-f001:**
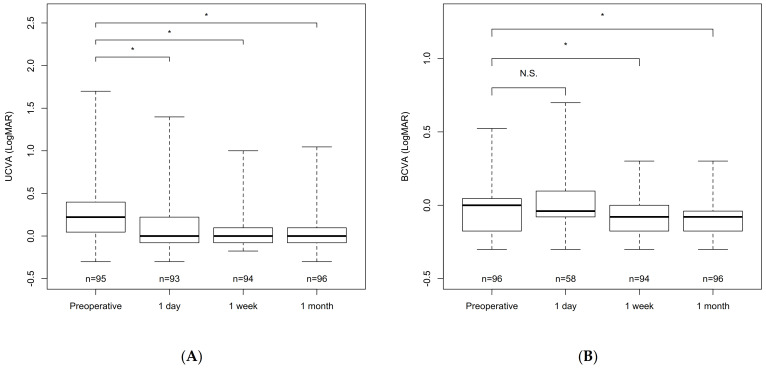
Box plots of each variable. (**A**) Changes in the uncorrected visual acuity (UCVA), (**B**) best-corrected visual acuity (BCVA), (**C**) spherical equivalent, (**D**) astigmatism, and (**E**) endothelial cell density. * indicates a significant difference between baseline and each time point. N.S., not significant; D, diopters; logMAR, logarithm of the minimum angle of resolution.

**Table 1 jcm-10-03672-t001:** Demographic data of the study participants.

Variable	Value
No. eyes	96 eyes/91 patients
Right/left eyes	48/48
Women/men	69/27
Age at cataract surgery (years)	55.0 ± 7.5
Age at pIOL implantation (years)	45.3 ± 7.4
Duration between surgeries (years)	9.7 ± 3.6
Emery-Little classification of nuclear cataract (eyes)	Grade I (16), grade II (46), grade III (31), grade IV (3)
pIOL material (eyes)	PMMA (53), silicone (43)
UCVA (logMAR)	0.29 ± 0.34
BCVA (logMAR)	−0.01 ± 0.17
Target refraction (D)	−0.17 ± 0.49
Spherical equivalent (D)	−1.43 ± 1.59
Cylinder (D)	−0.82 ± 0.73
Endothelial cell density (cells/mm^2^)	1986 ± 732
Axial length (mm)	28.39 ± 1.94

Values are expressed as the mean ± standard deviation. The date of previous pIOL implantation was unknown in seven eyes and the UCVA before the cataract surgery in one eye. The age at pIOL implantation and duration between the surgeries were calculated without these eyes and the UCVA without the one eye. pIOL, phakic intraocular lens; PMMA, polymethyl methacrylate; UCVA, uncorrected visual acuity; BCVA, best-corrected visual acuity; D, diopters.

**Table 2 jcm-10-03672-t002:** Changes in each variable.

	UCVA (logMAR)			BCVA (logMAR)			SE (D)			Cylinder (D)			ECD (cells/mm^2^)		
	*N*	Variable	*p* Value	*N*	Variable	*p* Value	*N*	Variable	*p* Value	*N*	Variable	*p* Value	*N*	Variable	*p* Value
Preoperative	95	0.29 ± 0.34		96	−0.01 ± 0.17		96	−1.43 ± 1.59		96	−0.82 ± 0.73		96	1986 ± 732	
Day 1	93	0.11 ± 0.30	<0.001 *	58	0.01 ± 0.17	1.0	58	−0.32 ± 1.12	<0.001 *	58	−1.33 ± 1.45	0.0060 *			
Week 1	94	0.06 ± 0.23	<0.001 *	94	−0.08 ± 0.12	<0.001 *	94	−0.18 ± 0.78	<0.001 *	94	−0.82 ± 0.97	1.0			
Month 1	96	0.03 ± 0.19	<0.001 *	96	−0.09 ± 0.10	<0.001 *	96	−0.17 ± 0.84	<0.001 *	96	−0.57 ± 0.59	0.078	96	1897 ± 725	<0.001 *

The values are expressed as the mean ± standard deviation. * Statistically significant difference between baseline and each time point. UCVA, uncorrected visual acuity; BCVA, best-corrected visual acuity; SE, spherical equivalent; ECD: endothelial cell density; D, diopters.

**Table 3 jcm-10-03672-t003:** Postoperative complications.

Complication	% (eyes)
IOP elevation exceeding 25 mmHg	10.4% (10)
Corneal edema	8.3% (8)
Iritis	3.1% (3)
Corneal epithelial defect	2.1% (2)
Hyphema	2.1% (2)

IOP, intraocular pressure.

## Data Availability

The data presented in this study are available on request from the corresponding author with the permission of the Keio University Ethics Committee. The data is stored, and it will be discarded after the approved period by Ethics Committee.

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
