# Peer review of "Short-Term Efficacy and Safety of Cataract Surgery Combined with Iris-Fixated Phakic Intraocular Lens Explantation: A Multicentre Study"

_jcm, 2021, doi:10.3390/jcm10163672_

Round 1

Reviewer 1 Report

Reviewer’s Comments:

The manuscript "Short-term efficacy and safety of cataract surgery combined with iris-fixated phakic intraocular lens explantation: a multicentre study" by Omoto et al   evaluated the short-term efficacy and safety of cataract surgery for 18 patients with iris-fixated phakic intraocular lenses (pIOLs).

  1. Was informed consent obtained from the patients?
  2. Please add a brief paragraph on “future directions to this study” at the end of the discussion/conclusions section.
  3. Please be consistent with the style of references.
  4. Please proofread for spelling and grammatical errors.

Author Response

Comment: Was informed consent obtained from the patients?

Response: All patients provided written informed consent for the surgery. As mentioned in the Informed Consent Statement, patient consent “for participating in this study” was waived and the opt-out approach was used according to the Ethical Guidelines for Medical and Health Research Involving Human Subjects presented by the Ministry of Education, Culture, Sports, Science and Technology in Japan.We edited the statement as follows: .“All patients read and signed the written informed consent form at each institute before the surgery. Patient consent for participating in this study was waived and the opt-out approach was used according to the Ethical Guidelines for Medical and Health Research Involving Human Subjects presented by the Ministry of Education, Culture, Sports, Science and Technology in Japan.”

Comment:Please add a brief paragraph on “future directions to this study” at the end of the discussion/conclusions section.

Response:We edited the text as follows: “Our follow-up period was short. Although the recovery from the surgery was favorable despite this short-follow up period, the information about the clinical outcomes and safety with a longer follow-up is essential for clinicians. In particular, the ECD significantly decreased at month 1 and careful follow-up is needed. This is also going to be discussed in the near future.”

Comment:Please be consistent with the style of references.

Response:We edited the reference style.

Comment:Please proofread for spelling and grammatical errors.

Response:An English-speaking editor reviewed the paper.

Reviewer 2 Report

This ms is very interesting. There are some questions and comments below;

1. The IOL power calculation methods should be described in the method. I think the iris-fixated phakic IOLs might influence the results of biometry (anterior chamber depth, axial length, etc.). Did you take such influence into consideration for IOL power calculation?

2. Was there any difference in visual outcomes between the eyes with PMMA phakic IOLs and those with silicone phakic IOLs? The induced astigmatisms resulting from the explantation were estimated to be larger in the PMMA group than in the silicone group.

3. In this study, the efficacy and safety indices (1.02 ± 0.56 and 1.31 ± 0.64, respectively) were referred to in the light of refractive correction. However, these indices might inappropriate for evaluating the results of cataract surgery because the preoperative visual acuity is affected by the severity of cataract. How do you think about it?

Author Response

Comment:1. The IOL power calculation methods should be described in the method. I think the iris-fixated phakic IOLs might influence the results of biometry (anterior chamber depth, axial length, etc.). Did you take such influence into consideration for IOL power calculation?

Response: We agree that pIOLs can affect the biometric measurements. However, to the best of our knowledge, there is no established method to adjust for this effect. We added the following sentence: “The IOL power was calculated using Barrett Universal II Formula with the preoperative measurements of axial length, keratometry, and anterior chamber depth. The anterior chamber depth was manually measured and checked for accuracy because the participants had pIOLs.”

Comment 2.Was there any difference in visual outcomes between the eyes with PMMA phakic IOLs and those with silicone phakic IOLs? The induced astigmatisms resulting from the explantation were estimated to be larger in the PMMA group than in the silicone group.

Response:We agree with you and performed an additional comparative analysis on the astigmatism between the eyes with PMMA and silicone pIOLs. As a result, the difference was significant only preoperatively. It is noteworthy that the astigmatism was subjective in the study and toric IOLs were inserted in some cases. We added this result to the manuscript as follows: “The preoperative subjective astigmatism was significantly different between the patients with PMMA pIOLs and those with silicone IOLs (p=0.022, supplemental figure 1). However, this difference was not found postoperatively. “

Comment 3.In this study, the efficacy and safety indices (1.02 ± 0.56 and 1.31 ± 0.64, respectively) were referred to in the light of refractive correction. However, these indices might inappropriate for evaluating the results of cataract surgery because the preoperative visual acuity is affected by the severity of cataract. How do you think about it?

Response:We agree that these indices were inappropriate for ordinary cataract surgeries; however, as mentioned in the manuscript, we included eyes without visual impairment in this study. In fact, the mean preoperative BCVA was better than 1.0 (logMAR -0.01) as shown in table 2. Therefore, we added the following sentence:  “We calculated these indices because the current surgeries reported in this study were performed on patients without visual impairment in many cases.”

Reviewer 3 Report

The authors present the results of a retrospective study of anterior pIOL removal and subsequent cataract surgery and in-the-bag IOL placement. As the authors mention, there is a paucity of this type of manuscripts, specifically considering the specific aspect of studying anterior chamber pIOL. The sample size is reasonable, and although the results are interesting, there are several aspects that should be clarified prior to consider its publication.

Material and methods

  • The main limitation of the study is the extremely short follow-up of the patients. A longer follow-up period could be very helpful as the potential readers could obtain further information about safety, ECC and refractive outcomes. Most of these reports, currently available literature, include data from at least 6 or 12 months postoperatively.[1]
  • Surgical technique: how many different surgeons participated in the surgery? It should be specified. The differences in the surgical technique may somehow influence in the postop results, especially since they are reported with a very short follow-up time
  • Surgical technique: range of the corneal wound to explant de anterior chamber pIOL was 2.4-7.0 Did the authors perform a scleral tunnel to explant the lens, or were the lens extracted via clear cornea? Also, up to 3.2 mm wide, it is impressive, as it was not mentioned in the manuscript, that the authors did not use nylon sutures to close the wound, which could totally influence the results. When were the sutures, if that were the case, removed?
  • IOL calculation: biometers are included, but not the formula used for IOL calculation. Although the measures are generally accurate,[2] some miscalculation in the axial length can be found and the subsequent hyperopic change.[3] It would be very interesting if the authors include this aspect. The authors mentioned they will include this information in future reports, but the reason of not including this in the present report remains undisclosed.
  • Statistics: there are other ways, and maybe simpler to perform, as Generalized Estimating Equations, to consider both eyes of the same patient

Results

  • More than the material itself, the authors included in the analysis several types of IOLs; furthermore, some of them are multifocal IOLs, and although it may not suppose a great limitation, this aspect has completely been overlooked.
  • This also applies to astigmatism: some of them are toric IOLs, some of them are not. Both aspects should be discussed.
  • Graphs showing visual acuity evolution (and other parameters included in the study) would help to improve the interpretation of the study.

Discussion

  • The discussion is correct, but indeed some of the aforementioned aspects should be taken into account. All of them must be included in the revised version and those that could not be addressed, should be included as important limitations of the study.

Bibliography

  1. Vargas V, Alió JL, Barraquer RI, et al (2020) Safety and visual outcomes following posterior chamber phakic intraocular lens bilensectomy. Eye Vis (London, England) 7:34. https://doi.org/10.1186/s40662-020-00200-8
  2. Amro M, Chanbour W, Arej N, Jarade E (2018) Third- and fourth-generation formulas for intraocular lens power calculation before and after phakic intraocular lens insertion in high myopia. J Cataract Refract Surg 44:1321–1325. https://doi.org/10.1016/j.jcrs.2018.07.053
  3. Yaşa D, Köse B, Sucu ME, Ağca A (2020) Intraocular lens power calculation in a posterior chamber phakic intraocular lens implanted eye. Int Ophthalmol 40:2017–2022. https://doi.org/10.1007/s10792-020-01377-6

Author Response

Comment: The main limitation of the study is the extremely short follow-up of the patients. A longer follow-up period could be very helpful as the potential readers could obtain further information about safety, ECC and refractive outcomes. Most of these reports, currently available literature, include data from at least 6 or 12 months postoperatively.[1]

Response: We agree with your comment and this is a limitation of our study. We reemphasized this point in the Discussion as follows: “Our follow-up period was short. Although the recovery from the surgery was favorable despite this short-follow up period, the information about the clinical outcomes and safety with longer follow-up is essential for clinicians. In particular, the ECD significantly decreased at month 1 and careful follow-up is needed. This is also going to be discussed in the near future.”

Comment: Surgical technique: how many different surgeons participated in the surgery? It should be specified. The differences in the surgical technique may somehow influence in the postop results, especially since they are reported with a very short follow-up time

Response: We added the following sentence: “Five surgeons performed all of the surgeries.”

Comment: Surgical technique: range of the corneal wound to explant de anterior chamber pIOL was 2.4-7.0 Did the authors perform a scleral tunnel to explant the lens, or were the lens extracted via clear cornea? Also, up to 3.2 mm wide, it is impressive, as it was not mentioned in the manuscript, that the authors did not use nylon sutures to close the wound, which could totally influence the results. When were the sutures, if that were the case, removed?

Response: As you indicated, the suture was set when the PMMA lens was explanted, which was left at the site until the end of the study period. We added this information to the surgical method section. We also re-analyzed the data and compared the astigmatism between the silicone and PMMA groups. As a result, the difference was significant only preoperatively. It is noteworthy that the astigmatism was subjective in the study and toric IOLs, as you indicated, were used in some cases. We also added this result to the manuscript: “The nylon suture was set when the PMMA lens was explanted, which was left at the site until the end of the study period.”

“The preoperative subjective astigmatism was significantly different between the patients with PMMA pIOLs and those with silicone IOLs (p=0.022, supplemental figure 1); however, this difference was not found postoperatively. “

Comment: IOL calculation: biometers are included, but not the formula used for IOL calculation. Although the measures are generally accurate,[2] some miscalculation in the axial length can be found and the subsequent hyperopic change.[3] It would be very interesting if the authors include this aspect. The authors mentioned they will include this information in future reports, but the reason of not including this in the present report remains undisclosed.

Response: Yes, the optimized formula for the surgery is really interesting, although it is regretful that the analysis has not been completed yet and we focused on the clinical results and the safety in the study. However, we agree that the information about the formula used in the cases was missing and the reports raised by you are interesting. We added the following text to the manuscript: “The IOL power was calculated using Barrett Universal II Formula with the preoperative measurements of axial length, keratometry, and anterior chamber depth. The anterior chamber depth was manually measured and verified for accuracy by the examiner because the participants had pIOLs.”

“Although it was reported that the preoperative biometric measures were generally accurate, (Amro M, Chanbour W, Arej N, Jarade E (2018) Third- and fourth-generation formulas for intraocular lens power calculation before and after phakic intraocular lens insertion in high myopia. J Cataract Refract Surg 44:1321–1325. https://doi.org/10.1016/j.jcrs.2018.07.053), some miscalculations in the axial length were found along with the subsequent hyperopic change. (Yaşa D, Köse B, Sucu ME, Ağca A (2020) Intraocular lens power calculation in a posterior chamber phakic intraocular lens implanted eye. Int Ophthalmol 40:2017–2022. https://doi.org/10.1007/s10792-020-01377-6)”

Comment: Statistics: there are other ways, and maybe simpler to perform, as Generalized Estimating Equations, to consider both eyes of the same patient

Response: We re-checked the p value with the mixed effect model and some were incorrect. We apologize for this error and amended table 2 and the manuscript. We agree that GEE is another appropriate method in this case. GEE and the mixed effect model differ, as you might know better than we do, depending on whether the target inference is the population-average (GEE) or subject-specific (mixed-effect model) but similar in that the measurements were nested within the groups, in this case, within the subjects (Hubbard AE et al. Epidemiology. 2010 Jul;21(4):467-74). We assume that the difference between the 2 models is marginal and the mixed effect model is still appropriate when dealing with both eyes as many other studies do. In fact, the p values calculated using these 2 models were similar as shown below. However, we will be happy to replace the results with those obtained with the proposed modeling if you prefer this.

UCVA

BCVA

SE

Cylinder

ECD

N

Variable

p value (GLM)

p value (GEE)

N

Variable

p value (GLM)

p value (GEE)

N

Variable

p value (GLM)

p value (GEE)

N

Variable

p value (GLM)

p value (GEE)

N

Variable

p value (GLM)

p value (GEE)

Preoperative

95

0.29 ±0.34

96

-0.01 ±0.17

96

-1.43 ±1.59

96

-0.82 ±0.73

96

1986 ±732

Day 1

93

0.11 ±0.30

< 0.001*

< 0.001*

58

0.01 ±0.17

1

0.81

58

-0.32 ±1.12

< 0.001*

< 0.001*

58

-1.33 ±1.45

0.0060*

0.021*

Week 1

94

0.06 ±0.23

< 0.001*

< 0.001*

94

-0.08 ±0.12

< 0.001*

< 0.001*

94

-0.18 ±0.78

< 0.001*

< 0.001*

94

-0.82 ±0.97

1

1

Month 1

96

0.03 ±0.19

< 0.001*

< 0.001*

96

-0.09 ±0.10

< 0.001*

< 0.001*

96

-0.17 ±0.84

< 0.001*

< 0.001*

96

-0.57 ±0.59

0.078

0.019*

96

1,897 ±725

< 0.001*

< 0.001*

Results

Comment: More than the material itself, the authors included in the analysis several types of IOLs; furthermore, some of them are multifocal IOLs, and although it may not suppose a great limitation, this aspect has completely been overlooked.

This also applies to astigmatism: some of them are toric IOLs, some of them are not. Both aspects should be discussed.

Response: The types of inserted lens varied and this is another limitation of our study, which we mentioned in the revision:

“Furthermore, the types of inserted lens varied and included toric and multifocal IOLs because of the multicentre study.”

Comment: Graphs showing visual acuity evolution (and other parameters included in the study) would help to improve the interpretation of the study.

Response: We submitted figures 1A, 1B, 1C, and 1D that show the changes in each parameter over time. We would appreciate if you could mention your specific suggestions regarding changes to the figures.

Discussion

Comment: The discussion is correct, but indeed some of the aforementioned aspects should be taken into account. All of them must be included in the revised version and those that could not be addressed, should be included as important limitations of the study.

Response: Because of the retrospective and multicentre design, the data are limited. However, we believe our large collection of surgical data is still valuable for the readers and wish this revision has successfully addressed the Review’s complementary suggestions.

Result after correction

Figure 1A and table 2show the changes in the UCVA. The preoperative value significantly improved at day 1, week 1, and month 1 postoperatively (p < 0.001 for all comparisons by a linear mixed-effect model followed by Dunnett’s test). Similarly, the postoperative BCVA improved significantly at week 1 and month 1 (p < 0.001 for both comparisons) but not on day 1 (p = 1.0, figure 1B, table 2).The efficacy and safety indices on postoperative month 1 were 1.02± 0.56 and 1.31 ± 0.64, respectively.

The subjective SE was significantly larger (closer to 0) at all time points (p < 0.001 for all comparisons) (figure 1C, table 2), whereas the subjective astigmatism was greater on day 1 (p = 0.0060) but did not differ significantly at week 1 and month 1(p = 1.0 and p = 0.078, respectively) (figure 1D, table 2). The ECD significantly decreased at month 1 (p < 0.001) (figure 1E, table 2).

Round 2

Reviewer 3 Report

The authors have significantly improved the manuscript. Some of the methodological concerns previously exposed were discussed and corrected, some of them were not.

There are some aspects, however, that are not entirely clear. The limitation paragraph is still missing, and the authors may consider including a more detailed discussion about the flaws of the present research. The retrospective design, the ECD loss, the reasons for not doing a longer follow-up since it was a retrospective study (missing data?) or the management/implications of the IOP spikes are found in the abstract but are not discussed any more.

Also, at the present version, the last paragraph and the supposed-to-be limitations paragraph are the same. Please correct that.

Author Response

Comment: The limitation paragraph is still missing, and the authors may consider including a more detailed discussion about the flaws of the present research. The retrospective design, the ECD loss, the reasons for not doing a longer follow-up since it was a retrospective study (missing data?) or the management/implications of the IOP spikes are found in the abstract but are not discussed any more.

Response: Thank you for your comment. The discussion about ECD loss was now added in the manuscript. Other limitations of our study are discussed in page 9 line 196 to page 9 line 201 (the paragraphs starting with “The current study had some limitations, one of which was “). The discussion about IOP spikes is in the second paragraph (page 9 lines 185 to page 9 line 195).

“In particular, the ECD significantly decreased at month. The explantation of pIOL was carefully performed through sclerocorneal incision not to touch the endothelium and this procedure specific to the surgery might have had affect. However, the ordinary cataract surgery with phacoemulsification and IOL implantation is well known to have effect on ECD. The ECD change in our study was in average 4.5 % and this was comparable to the past study of ordinary cataract surgery. (Hayashi K, Yoshida M, Manabe S, Hirata A. Cataract surgery in eyes with low corneal endothelial cell density. J Cataract Refract Surg. 2011 Aug;37(8):1419-25. doi: 10.1016/j.jcrs.2011.02.025. Epub 2011 Jun 17. PMID: 21684110.)(Goles N, Nerancic M, Konjik S, Pajic-Eggspuehler B, Pajic B, Cvejic Z. Phacoemulsification and IOL-Implantation without Using Viscoelastics: Combined Modeling of Thermo Fluid Dynamics, Clinical Outcomes, and Endothelial Cell Density. Sensors (Basel). 2021 Mar 30;21(7):2399. doi: 10.3390/s21072399. PMID: 33808502; PMCID: PMC8037460.) Thus, the ECD change was relatively small but the early endothelial cell change cannot be fully evaluated by ECD. (Kim DH, Wee WR, Hyon JY. The pattern of early corneal endothelial cell recovery following cataract surgery: cellular migration or enlargement? Graefes Arch Clin Exp Ophthalmol. 2015 Dec;253(12):2211-6. doi: 10.1007/s00417-015-3100-5. Epub 2015 Jul 14. PMID: 26170045.) Although the number of cases will be limited due to the retrospective design, careful and longer follow-up is needed, and this is also going to be discussed in near future.”

Comment: Also, at the present version, the last paragraph and the supposed-to-be limitations paragraph are the same. Please correct that.

Response: Thank you for your indication. The last paragraph was incorrectly changed in the last revision and now it was reverted to the previous version.

“In conclusion, cataract surgeries for patients with iris-fixated pIOL were performed safely with good visual outcomes. We believe this option may be considered for patients with a pIOL who have visual impairment and endothelial cell loss.”
